# Where Are We Now? Feeds, Feeding Systems and Current Knowledge of UK Horse Owners When Feeding Haylage to Their Horses

**DOI:** 10.3390/ani13081280

**Published:** 2023-04-07

**Authors:** Meriel Moore-Colyer, Amy Westacott, Lucile Rousson, Patricia Harris, Simon Daniels

**Affiliations:** 1School of Equine Management and Science, Royal Agricultural University, Stroud Road, Cirencester GL7 6JS, UK; 2AgroSup, Rue St Pettion, 21000 Dijon, France; 3Equine Studies Group, Waltham Petcare Science Institute, Waltham-on-the Wolds, Leicestershire LE14 4RT, UK

**Keywords:** horses, forage, haylage, hay, feeding, diet, owner education

## Abstract

**Simple Summary:**

Despite our improved understanding of equid digestive health and new feeding systems, the level of obesity in the UK horse population remains high. This study aimed to determine how owners are feeding their horses, what influences feeding practices, owner’s knowledge of haylage and what areas of feeding require supplementary education. Data were collected in 2020 from 1338 UK horse owners via two online surveys. Survey 1 was on general feeding practices, while Survey 2 was specifically on feeding haylage. Equal numbers of leisure and performance horse owners completed both surveys. In Survey 1, 67% were fed hay as the only forage, 30% were fed forage (hay/haylage) + balancer, 36% were fed both haylage and hay that was done for managing the energy intake and 84% added a cube or coarse mix feed; forage analyses were uncommon, as 88% had no analyses: 74% did not see the need for it and 16% did not know analyses could be done. Of those (Survey 2) who were not feeding haylage, 66% said they were not sure how to feed it, 68% worried about aerobic spoilage and 79% said the bale size was unsuitable. Body weight measurements, as well as feed analyses (Survey 1 and Survey 2) were rarely performed (11%). Aspects of ration formulation, the value of feed analyses and how to interchange hay and haylage require additional education to owners if diets for horses are to be improved.

**Abstract:**

Despite our improved understanding of equid digestive health and accurate rations formulations, obesity in the UK horse population remains high. Study aims: (1) to determine how owners are feeding their horses and what influences their choices, (2) to understand owners’ knowledge of haylage and (3) to identify key areas that require additional education. Data were collected in 2020 from 1338 UK horse owners via two online surveys. Survey 1 was on general feeding practices, and Survey 2 was specifically on the feeding of haylage. Data were processed using chi square analyses + Bonferroni tests, with a significance *p* < 0.05. Equal numbers of leisure and performance horse owners completed both surveys. For Survey 1, 67% fed hay as the only forage, 30% fed forage (hay/haylage) + balancer, 36% fed haylage and hay to manage energy intake, 84% added a cube or coarse mix, 88% did not do forage analyses, 74% did not see the need for it and 16% did not know analyses could be done. In Survey 2, those who were not feeding haylage, 66% were not sure how to feed it, 68% worried about aerobic spoilage and 79% said the bale size was unsuitable. Body weight measurements (Survey 1 and Survey 2) were rarely performed (11%). Aspects of ration formulations, the value of feed analyses and how to interchange hay and haylage require additional education to owners for improved ration compilation.

## 1. Introduction

Equine nutrition research over the last 25 years has greatly improved our knowledge of equid digestive physiology and diet formulation for horses [1]. Investigations into the use of all fibre diets and the influence that different fibre types, conservation techniques and feeding systems have had on the dietary energy supply have been well reported [2,3,4,5,6,7,8]. Digestive and mental health issues have also received attention [9]. However, despite all these advances, the prevalence of obesity in the UK equid population remains very high, indicating a mismatch between energy intake (and, therefore, dietary choice) and energy requirements [10,11,12].

In a 1006-strong survey by King in 2012, hay + a cereal-based complementary feed was still the diet of choice of most UK horse owners [13]. However, in certain circumstances, such diets can be associated with excessive energy intakes and adverse clinical and behavioural issues [4,14]. To improve digestive health and time budgets in stabled horses, diets low in cereals and rich in fibre are now often recommended by nutritionists when compiling rations for horses across a wide range of activities [15]. Whilst an increasing number of commercial complementary feeds (i.e., feeds designed to be fed in combination with fresh or preserved forage) are available that are low in energy and starch but high in fibre, others are high in oil and highly digestible fibre sources [16,17]. The latter makes them good replacements for cereal-based concentrates when energy demands are high but can contribute to excessive energy intakes if fed to animals with low requirements or when used to complement high energy-providing forages [16,17].

Most UK horse owners appreciate that forage, i.e., grass/hay/haylage, should comprise at least 50% of the daily diet to ensure good gut health and natural time budgets, and it has been recommended that forage should be provided to most horses and ponies at a minimum of 1.5% body weight on a dry matter (DM) basis/day [4]. UK hay is typically cut and conserved at a mature physiological age and may not contain the nutritional profile an individual horse requires [15,18,19]. Choosing a complementary feed to supplement and balance the ration can be difficult, particularly for leisure horse owners and those with horses doing light work, as achieving trickle feeding and natural time budgets without overfeeding energy can be challenging. These issues are also a challenge for those owners choosing to feed hay as a means of managing excessive weight gain in their horses. The protein and mineral levels can be low in very mature fibrous forage, and balancing these nutrients needs to be considered, especially for a horse on a strict weight loss regimen [15]. The increasing availability of forage balancers (concentrated supplementary feeds formulated typically to supply quality protein, vitamins and minerals in small amounts without adding substantially to the energy intake) as an alternative to more energy-rich complementary feeds is a potential solution to this issue.

With the challenges associated with conserving good quality hay, many owners now choose haylage as the main forage for their horses. Haylage can be put into two broad categories: the first is generally produced from less mature and more energy-rich grass, has undergone some fermentation and has a high proportion of leaf: stem and a DM range between 50 and 70% [4]. The second category is effectively wrapped mature grass and has a DM and nutritional profile closer to hay with minimal fermentation. Haylage is a relatively new feed for horses, and the lack of information surrounding the appropriate conservation, nutritional values, aerobic stability and its interchangeability with hay poses a challenge for many owners. Anecdotal evidence suggests that owners who do not use haylage base this on their perception that haylage is too acidic, too energy-dense and unsuitable for horses with gastric ulcers. These reasons may indeed be accurate for their horse and the haylages they have encountered but should not be generally regarded as proven characteristics of all haylages. For some horse owners, utilising haylage has been shown to support high-level activity, as demonstrated by standardbred trotters, which performed equally well on a diet based on haylage (with a fortified balancer) compared with a hay and cereal-rich diet [20,21]. Furthermore, haylage can be an excellent feed for young, fast-growing thoroughbreds that achieved equal growth rates compared to those fed high levels of cereals (up to 6 kg/day) [22]. While the advantages of high nutritive value haylage are clearly evident, haylage can be a very variable forage, depending on the physiological age of the grass, grass species, conservation time and edaphic conditions, resulting in great variability in terms of the DM content and nutritional profile, as discussed above [4].

In order to determine what the owners know about haylage and their feeding decisions, as well as their current feeding practices, the extent their feeding practices have changed over the last decade and to see if recent research findings are being applied to the practical feeding of horses, this study aimed to:Determine current feeding habits of the UK horse-owning public (Appendix A).Determine owner knowledge and attitude when feeding haylage (Appendix A).Highlight areas that require knowledge dissemination and identify topics for further research that are most relevant to the horse-owning public.

## 2. Methods and Materials

This study consisted of two surveys of the horse-owning population across the UK and Ireland. Both surveys were advertised via social media and were available through the Joint Information Systems Committee (JISC) online and were ‘live’ for a designated time. The surveys contained the following statement: ’The survey will take you approximately 10 min to complete. Your participation in this study is entirely voluntary, all data are totally anonymous therefore once the questionnaire has been sent data cannot be identified to be removed. By submitting your results, you are therefore agreeing to the use of your data in this study.’ This survey was approved by the Royal Agricultural University Research Ethics Committee (project approval number: 2020.0040).

### 2.1. Survey Design

Survey 1 (S1) was circulated between November 2019 and February 2020 and was available for completion for 73 days. It was promoted to universities, riding clubs and pony clubs and via Facebook groups. The survey consisted of 21 main questions that gathered data on horse and owner demographics and knowledge, general feeding practices and the management of different feed types. The questions were a mixture of 11 single answer questions, 17 multiple choice questions, for which respondents could select more than one answer, and 2 ranking questions. This meant that there was an overall ‘response effort’ of thirty-one questions.

Survey 2 was circulated from June 2020 to September 2020 and was ‘live’ for 120 days. It was distributed via social media as well as horse and rider groups over the UK and via online forums (Horse and Hound). This survey was primarily targeted at owners who used haylage, and it was titled ‘Haylage survey’. However, there was a section on the knowledge of haylage that could be completed by owners that did not use haylage, with the objective of determining the reasons behind not using this type of conserved forage. Although the questions were wide-ranging and covered owners’ perceptions and knowledge about haylage, including handling, cost, source, bale size, quality aspects, suitability for their horses and health issues thought to be associated with haylage, it also posed questions regarding feeding management, factors that influenced forage choices, feed analyses and if owners sought any advice about feeding and from whom. Thus, the data presented from this survey arose from 15 main questions, and as some were ‘funnelling’ questions, the overall response effort by some respondents was, therefore, 113 questions.

### 2.2. Data Analyses

The surveys were analysed separately, but both were subjected to the following procedures. Data were exported into an Excel spreadsheet, where ‘spoiled’ surveys, identified on the basis of contradictory information, e.g., ticking multiple boxes or filling in more horses than were initially recorded, were removed before the analyses. The questions were analysed based on the number of responses to that section/question, rather than automatically applying the entire number of survey respondents. Questions with yes/no answers and those ranking information via a scale were analysed using the chi square and Bonferroni correction tests to determine significant differences with significance taken as equal to or lower than *p* < 0.05 (Genstat 19). The data in S1 were generally divided into two groups: 15–40 (G1) and 41+ years old (G2). This division was based on the likelihood that longer-term experience and tradition might have influenced the management system the owner adopted and were analysed by multiple chi square analyses. Some questions, where age was unlikely to affect management choice, were put into a single group and analysed using a single group chi square test.

For S2, where appropriate, single group chi square analyses were performed, and Bonferroni correction tests were used to determine significant differences. Significance was taken as equal to or less than *p* < 0.05 (Genstat 19).

## 3. Results

There were 504 responses used for analysis from S1 and 834 for S2. Of the 834 total responses to S2, 73.5%, i.e., 613 people, fed haylage to their horses, while 221 (26.5%) did not feed haylage. Answers about haylage quality could not be answered by those who did not feed haylage, so the results presented on quality were from 613 owners that had experience feeding this forage.

### 3.1. Respondent Demographics

Survey 1 respondents were divided into two age categories, which resulted in a similar number of individuals per group. GI aged between 15 and 40 years comprised 290 people, and G2 aged between 41 and 50 years comprised 214 individuals. Survey 2 did not categorise respondents by age.

### 3.2. Riding Activity

Across all 504 respondents in S1, 41% were self-identified as pleasure riders, with the remaining 34% divided into roughly 10% in each of the following categories of eventing, show jumping and dressage, totalling 75% of respondents. The remaining 25% were divided between hunting, showing, Western riding and driving.

There was a significant difference between the age groups in the type of education they had received, with 118 out of 294 of G1 having a university-level education (e.g. BSc/MSc) compared with only 28 out of 210 in G2. Similar numbers were noted in both groups with regard to formal or industry-level education (see Table 1).

The total number of horses and ponies owned by the 504 respondents was 735, and the majority (535) of these were horses, i.e., >14.2 hands high.

S2 did not ask directly about owners’ education, the number of horses owned or what activity they did.

### 3.3. Management Regimen

Survey 1 revealed that both age groups of owners managed their horses similarly (*p* < 0.060), with both groups using a variety of regimens, as shown in Table 2. Significantly more, i.e., 328 owners, preferred a mixture of stabling and turn out compared with the other two regimens (*p* < 0.001). Significantly more owners never stabled their horses compared with those who had their horses continually indoors (*p* < 0.003).

The stable management regimens detailed in S2 showed a similar trend, although the percentages were a little different. The majority used a mixed regimen of stabling interspersed with turn out, with 79.7% doing this during the winter, 69% in the spring and autumn and 47% in the summer. This was further broken down into 97% stabling for most of the time in the winter, 27% stabling for most of the time in the spring and autumn and only 7% stabling for most of the time during the summer.

Very few, only 31.5%, kept their horses constantly out at grass during the winter. This number increased during the drier seasons, with 65% of owners keeping horses out during the spring and autumn and 96.8% out all the time during the summer.

### 3.4. Exercise

Survey 1 revealed that 72.1% of owners rode their horses more than 3×/week, with 19.5% riding twice/week, while 8.4% of horses were identified as retired and so were not ridden at all.

### 3.5. Forage Choice and Feeding Frequency

The results from S1 on the forage feeding regimen were analysed as one group, and 56% of owners fed their horses forage twice/day, which was significantly more than the 30% who fed once per day and more than the 7% who fed three or more times/day (*p* < 0.006). Some owners, i.e., 5% of the 504 people to complete S1, did not feed their horses any complementary feed (*p* < 0.006), while 30% said they just fed forage and a balancer, while all others fed a complementary feed such as a commercial chopped fibre-based or a commercial coarse mix or cube/nut feed.

The type of forage fed was not influenced by the age group, with 328 (65.7%) feeding hay and only 76 (15.1%) feeding haylage, while 96 (19.2%) were feeding hay and another forage.

Most, i.e., 85%, did not weigh out the forage hay or haylage before feeding. Of the 78 who did weigh their forage, 45 owners fed between 5 and 10 kg of forage/day, 19 fed 10+ kg and 14 fed less than 5 kg.

Of the 834 horse owners that completed S2, 73.5% of them fed haylage, i.e., 613 horse owners. The reasons stated for choosing haylage over hay were to do with dust-related issues in hay. It is not clear how owners measured the dust levels, but when assessing the quality of the forage, they used other informal (nonproven) parameters, e.g., had lots of leaves (64.2%), quite moist (63%), no mould (61%), sweet smell (59%), acidic smell (43.4%), no dust (40.5%) and soft to the touch (40.6%).

The 221 (26.5%) that did not use feed haylage stated that they felt it was too high in energy and horses got fat (33% each), while 15.3% said their horses became too excitable on haylage. Other reasons for not feeding haylage were more practical, such as storage problems (24.4%), cost (22.2%), availability (13.6%) and bale size (79%). A group of owners, 27% of the 221, said they just preferred to feed hay. Additionally, there was a free text question that gave owners who did not feed haylage an option to explain why, and this resulted in a plethora of reasons, but the most common identified by this group were that haylage was too acidic and caused digestive upsets, caused loose droppings and many felt it was not suitable for horses with gastric ulcers (81%). Sixty-eight percent were concerned about dental problems, and ninety percent indicated it was not suitable for laminitics; 66% were not sure how to feed haylage, and 68% were nervous about aerobic spoilage.

When feeding both hay and haylage, 23% fed using a 50:50 ratio, while the rest fed a variety of different proportions of both forages. A notable proportion, i.e., 39.2%, said they fed both forages to give their horses variety in their diets, and 36% fed a variety of ratios as a way to manage the energy intake and weight loss. The rest (39.5%) fed a mixture of both forages for a plethora of other reasons, including cost, storage, access to forage and difficulty disposing of large quantities of plastic wrapping.

### 3.6. Owner Knowledge of Forage Nutritional Value

The knowledge on haylage from S2 revealed that 30% thought haylage was wrapped partially fermented grass, while 41% felt it was wrapped partially dried grass, and 91% said haylage had lower dust than hay, 52% said it had a higher nutritional content than hay, 43% said haylage was higher in water-soluble carbohydrates (WSC) than hay and 20% said it was more acidic than hay.

In S2, the owners could choose multiple answers, so some, 73%, preferred to feed just haylage due to its higher energy/nutrient content and to promote weight gain in their horses. Nearly a quarter of owners (21%) felt that hay and haylage were fully interchangeable and did not alter the weight fed according to forage type. Others, 26%, fed less haylage than hay, and 53% fed more haylage than hay. For those owners that fed a greater weight of haylage than hay, 94.2% said they did this because haylage had a higher moisture content than hay.

The majority of owners in S1, i.e., 89% (447), did not have their forage analysed, while only 57, i.e., 11%, did. Of the 89%,most owners, i.e., 74% of them, did not see the need for forage analyses, 9% said it was too costly and 16% did not know you could have forage analysed.

The results from S2 showed that a high proportion of owners, 73.7%, said their haylage did not come with any nutritional analyses. Some owners, 24.8%, reported that an analysis came from the farmer or, if they purchased commercially produced haylage from a recognised brand, the nutrient profile was on the pack. For those owners who paid for an analysis, 75.4% said they received fibre, protein and WSC values, but only 66.7% received the DM content. For those whose haylage did not come with any analyses, 95.8% did not subsequently get any analyses done.

Of the 29 (4.2%) respondents that did pay for an analysis, only 19.2% (5.5 owners) analysed each batch they received from same farm while 57.7% (16.7 owners) only did analyses on one batch per year, indicating that they assumed all haylage from that source had a similar nutrient profile. When a haylage analysis was sought, it was done through a feed company or a professional laboratory.

When receiving or seeking analyses, only 15.4% received a hygienic analysis. Half the hygienic analyses reported values for the dust content and weed contamination; whereas the other half received results for bacteria, and a quarter of these received a mould contamination report.

### 3.7. Feeding Regimens

More than half of the respondents in S1, 291 (58%), fed forage from a net, 162 (32%) fed from the floor and 51 (10%) fed from a rack.

A total of 465 owners fed a fibre-based bucket feed (chop) to their horses, while 39 owners did not feed any commercial chopped fibre feed, i.e., Alfalfa, chaff or sugar beet pulp. The type of chopped fibre feed varied, with 69% feeding a straw-based chaff, which was more than the 22% that fed alfalfa-based chaff and the 9% that fed sugar beet pulp (*p* < 0.003).

When feeding a complementary chopped fibre feed (i.e., defined in the question as a feed of >18% crude fibre), 75% measured out the amount to be fed, while 25% did not. When measuring out the complementary feed, 91% used a scoop or other container and did not use scales to accurately weigh out the feed.

In S1, 418 out of 504 respondents, i.e., 83%, fed a complementary feed to their horses. The type of feed was evenly distributed between the 134 who fed a cereal-based coarse mix and the 159 who fed nuts/cubes. Measuring out complementary feed was done by 359 (71%) owners, but 12% (59) did not weigh anything. As with the fibre feeds, most, i.e., 249 (70%), used some type of receptacle to measure out the feed, with 110 (31%) actually using scales.

Most owners in both G1 and G2 were consistent with the brand of forage/fibre-rich feed they fed to their horses, as detailed in Table 3, and the cited reason for using a particular brand was that it ‘worked well for their horses’ (*p* < 0.001). However, neither group were particularly brand loyal when feeding complementary feeds (Table 4).

### 3.8. Knowledge and Advice

A quarter (26%) of owners did not seek formal nutritional advice from nutritionists or vets prior to purchasing any form of horse feed.

Seventy percent of owners completing S1 fed supplements such as micronutrient-rich products (vitamin and mineral compounds) and nutraceutical dietary supplements for joint care. Of the 352 owners to give a reason why they fed a supplement, 60% fed them as a prevention or to balance the diet, whereas significantly less (132) owners fed supplements based on a diagnosed health condition (*p* < 0.036).

Of the 287 owners (74%) who sought advice before feeding supplements, 227 sought formal advice from a vet or nutritionist, while 60 acted on advice from friends, family, yard owners and internet forums.

As shown in Table 5, 42% of the G1 owners felt their feeding practices had changed over the last 5 years, and 43% said they had not changed in the last 10 years. The changes in feeding practices were attributed to more education/research being available, different horses needs and a wider range of feeds to choose from. Cost, time spent with their horse and the convenience of attaining a feed were ranked very low and, thus, were not catalysts for change over the last 10 years. A proportion of G2 respondents (31.8%) had changed their feeding practices in the last 5 years, but 35% had not changed anything at all And 33.2% had altered their feeding in the last 5–10 years. 

## 4. Discussion

### 4.1. Response Profile, Demographics and General Management Regimen

The responses from across both surveys came from 1338 horse owners and covered all regions of the UK. This number should make the results a robust example of the current feeding habits of UK horse owners and is similar in size to the studies of Agar et al., and Cameron et al., and notably more than the studies of Cerqueira et al., and Murray et al. [23,24,25,26]. The profiles of horse owners in S1 were a fairly even split between leisure horse owners (45%) and those that do more intense activities (therefore describing the feeding of performance horses) such as eventing, show jumping, dressage and hunting, so the results from this survey reflected management practices across a range of horse sports disciplines.

Although S1 focussed on general feeding management and education and S2 primarily examined owners’ attitudes and habits to feeding one particular type of forage, i.e., haylage, both surveys gave an interesting insight into the owners’ knowledge, highlighted some current issues and demonstrated the wide variety of management systems UK horse owners adopt when feeding their horses. The results also highlighted areas where the dietary health and weight management of horses could be improved if the nutritional profiles of forages were readily available and owners educated on how to interpret them.

As might have been expected, the level of formal education of the younger G1 group was significantly higher than in G2. This likely reflects the rise in the number of diploma and degree courses offered at UK higher education institutions over the last 15 years, which people over 40 would have had limited access to when choosing formal tertiary education. However, this does not seem to have influenced feeding choices, as answers to what was fed, management systems and knowledge of feed were similar between the groups. As survey responses cited sensible feed choices and feeding regimens, the parity between age groups suggests that other, less formal ways of knowledge exchange, e.g., lectures and talks via commercial feed companies, riding and pony clubs, online courses, websites and advertising, are effective conduits of up-to-date information on feeding and management practices.

The fact that a high proportion of owners (65%) kept their horses in a combination of turn out and stabling and 26% never stabled their horses agree with previous works and is still the most common management system used [27]. The deeper dive into the daily regimen in S2 revealed that, as expected, the season and climate played a major role, as was reported by Harris [28]. Most owners (79.7%) maintained a mixed regimen of stabling and turn out throughout the year, but the level of ‘mostly stabling’ increased during the winter months and decreased during the summer.

S2 did not specifically ask about what type of activity owners did with their horses, so the number of horses stabled throughout the season could be due to a range of reasons, such as limited turn out or competition preparation or simply a desire to control grass intake. However, it is good to note that, in both surveys, most horses still had some ‘field downtime’ each day, a beneficial practice for maintaining gut, limb and mental health [29].

### 4.2. Diet Compilation and Feeding Management

Most owners in S2 were self-confessed leisure riders and the majority (91%) rode only two to three times per week. Eighty-three percent fed a complementary feed (coarse mixes or compound nuts), which was generally given in two meals/day. The surveys did not identify how much feed was being fed and the exact content of the diet, nor the exact energy requirements for each individual horse or their body weights, but given the persistent high levels of obesity in the UK equine population, it is perhaps concerning that such a high proportion of our surveyed owners were feeding more than just a balancer (or low energy-fortified fibre-based feed) with their fresh or preserved forage [10,11,12,30]. More in depth investigations are required to explore the reasons behind this.

Excessive weight gain is obviously due to a mismatch between energy intake and energy expenditure, and this can occur both at grass and when stabled. While a high intake of grass is a common cause of weight gain in horses kept at pasture from the spring through to autumn, obesity is also seen in many stabled horses and can be attributed to the oversupply of high-energy complementary feeds and/or forages [31]. It is also worth noting that, when stabled horses are just standing, eating and not moving around to forage, their energy expenditure will be lower than when continually moving during grazing [32]. In the field study of Ebert and Moore-Colyer based on 175 recorded training sessions over 45 weeks, the commonly used system in the UK, i.e., NRC (2007) recommendations, were found to overestimate the energy requirements for mild exercise by 11% [33,34]. Coupled with this is the challenge faced by many owners of how to accurately determine the exercise level of their horses [35]. This means that, whether owners feed by eye, i.e., guesstimate, or apply any of the official recommendations, all too often they feed higher energy intakes than is actually required. The rising obesity rates demonstrate that owners are not practising objective body weight assessments and are therefore often unaware of the weight gain in their horses. This is a situation that needs immediate action, i.e., wide dissemination of more up-to-date, accurate and easy-to-use energy determination systems. Weight gain is a gradual process, so it is hard for owners to notice their horse is gaining weight without using more objective assessments, such as body condition scoring and weighing horses. Both measurements are easy to do and should be actively encouraged, but currently, the necessity of this sort of monitoring does not seem to be either widely recognised or taken up as a routine horse management activity. This potentially should be an ‘easy win’ against rising obesity and nutrition-induced metabolic disorders. However, again, the reasons why these procedures are not being more commonly utilised needs to be explored in more detail. Most yards do not have weighbridges, so a lack of access could be an issue. However, there are now increasing numbers of equine nutritionists who have mobile weighbridges, and so, obtaining an accurate weight for any horse is available and affordable for most owners. Failing this, weigh tapes, although not an accurate means of obtaining body weight, can be used to monitor changes in shape, provided it is placed correctly on each repetition [36]. In addition, nutritionists and veterinarians could actively promote and demonstrate simple energy requirement calculations, so that requirements are based on the individual horse’s actual energy expenditure, a value that is easily determined using a heartrate monitor and the user-friendly equation of Ebert and Moore-Colyer [33].

### 4.3. Knowledge on Feed Nutrient Content Analyses

Even if the owner can more accurately estimate the energy requirements of their individual animal, in order to correctly balance the energy intake and output, they obviously need to know the energy content of their forage, as this should be the basis of the diet and need to have determined the amount of forage their individual animal eats. The latter is more easily measured in animals that are stabled full time or only have access to a pasture for a very short period, but continues to be a challenge for those animals that are predominantly out to pasture [14]. This highlights the continued need to monitor the body weight and condition and for owners to adjust feed and forage types accordingly.

Many owners prefer to give at least one bucket feed per day, and this action can be stimulated by wanting to improve the horse–human relationship, such as entice an animal in from the field or to give a reward. Such desires can be largely met by making the correct choice of forage and fibre feed or balancer, which should be based on the requirements of each individual horse. However, making the correct choice is challenging when most forages are not accompanied by analyses. Without understanding the energy content of the feedstuff that comprises the majority of the ration, it is difficult to correctly choose the appropriate forage for an individual horse’s energy requirements, as discussed above. Although it is possible to meet the energy requirements of even high-performance sport horses on some forages alone, forages can have a wide variety of nutrient deficiencies, depending when, where and how they were conserved. Generally, most Northern Hemisphere hay has a low content of poor biological value protein [4]. Trying to assess the forage quality without doing a chemical analysis is flawed, as Julliand et al., reported that owners frequently deemed hay to be good quality when it had a high energy content, which might be suitable for harder working horses but not necessarily good for horses with low energy demands [14,37]. Moreover, variable mineral and micronutrient contents, influenced by varying edaphic conditions, fertilisers applications and harvesting conditions, can only be determined by proper analyses [38]. It could be argued that, in addition to analysing the dry matter and energy content, knowing the protein content should be the next step, as this could direct the choice of fortification towards either a protein-supplying balancer or a simple vitamin and mineral mix. However, importantly, the recommendation for obese horses and ponies and, in particular, those with insulin dysregulation and an increased risk of laminitis, is to feed (often low energy-providing) forage with a non-structural carbohydrate (NSC: starch and water-soluble carbohydrate) content of less than 12 or even 10% on a dry matter basis [15,36]. The physiological age, species profile and conservation timing (dry vs. wet) of haylage will indeed influence the WSC, which is crucially important for horses prone to laminitis. However, it is not possible to determine the WSC or NSC or any other nutrient content by eye, and therefore, an analysis is essential to accurately balance the ration. The majority of UK forages do require some level of supplementation, i.e., a balancer, and while feeding a fortified low-energy fibre feed is better than forage alone, it is preferable to choose a balancer from a position of knowledge of the nutrient profile of the basic diet. The majority of owners (74%) in this survey did not see the need for forage analyses, and 16% said it was too expensive. These results are supported by Cerqueira et al., who reported that 89% of owners from an 80-strong survey of horse owners in Florida did not have their hay analysed. Instead, most used a visual assessment [25,39].

Not knowing the nutritional profile of the forage is further compounded by the fact that results from the current surveys reported that most owners weighed out the complementary feed but did not measure the amount of forage they fed. It seems that weighing out forage was only considered when trying to control the body weight. Of the 78 respondents in S1 that did weigh the forage, 45 fed between 5 and 10 kg/day, and 14 fed less than 5 kg/day, which was undoubtedly influenced by the body weight and time spent at pasture. Although owners did not state if they fed forage ad libitum, which would be the system of choice, provided obesity is not an issue, as it provides constant access to feed, it is still necessary to have an idea of how much a horse is eating, so diets can be balanced properly. Not knowing the nutritional profile of the major component of the diet, nor how much the horse eats per day, makes matching the daily nutrient intake to the requirements almost impossible. Such feeding systems need a serious education campaign to:

(A) Encourage forage producers to have the feed analysed so owners know what they are buying.

(B) Inform owners of the importance of knowing the nutritional profile of forage.

(C) Having a clear idea of how much forage a horse eats per day when stabled.

(D) Choosing the appropriate complementary feed based on the nutritional profile of the forage and the individual’s requirements.

If all these recommendations were followed, many horses would benefit, feeding costs could be reduced, and it is highly likely that obesity and metabolic disorders would decrease.

### 4.4. Forage Choices

With respect to forage choice, S1 did show that, over the last 30 years, forage preference has remained in favour of feeding hay, and this choice was not influenced by age category, so was not a tradition-driven choice. In a survey reported by Harris, only 3% of the 86% of respondents that fed conserved forage fed haylage [28]. While the number of haylage users has increased a little since then, grass hay is still the preferred long-forage feed, with 69.3% of owners feeding hay over the winter period [13,40]. Of those owners who did not feed haylage, i.e., 26.5%, the main reasons for not doing so were around energy-related concerns such as weight gain and excitability. The health/medical reasons they cited were all related to acidity and the potential for digestive upsets. This perception has not altered much since 1997, when owners’ concerns over feeding haylage centred around the potential clostridial activity and lowered overall fibre intakes compared with hay [28]. These concerns could indeed be valid if the choice of haylage fed is not ideal for that horse, but such parameters need to be categorised and validated, so decisions are based on fact rather than assumption. This further strengthens the need for all forages, including haylage, to carry an accurate nutritional analysis, so those owners who choose to feed haylage for reasons of cost, availability and dust reduction measures, can determine which haylage would be best for their horse. The fact that S2 revealed many owners were feeding both hay and haylage and felt they were fully interchangeable while others fed either more or less compared with hay, indicates a dearth of information available on this forage. With the high level of variability of DM, acidity and nutritional profile of UK haylages, the only way to utilise this forage to its maximum potential is to have it analysed, so feed choices can be based on the nutrient content and daily requirements and not on just what is readily available [4].

Finally, when seeking nutritional advice, 26% of owners stated they did not seek any formal advice from qualified personnel such as nutritionists or vets and instead took advice from friends or sought help from the internet. Such strategies may initially seem more cost-effective but can result in a long process of trial and error and a yo-yo effect of weight gain and weight loss and the potential for poor welfare outcomes, which make this a false-economic choice. Owners should also be made aware that once a diet is accurately compiled, they should monitor the body weight and performance of their horses, as individual and feed preferences can influence the efficacy of a particular diet for an individual horse.

## 5. Conclusions

This study has shown that the more traditional diet of hay plus complementary feed (mainly commercial coarse mixes and compound nuts/cubes) is still the most common way UK horses are fed. Both surveys demonstrated that knowledge on certain aspects of feeding management such as access to pasture through the daily turn out and keeping variety in the diet are widely practiced, while other areas, e.g., appreciation of the value of forage analyses, knowing the nutritional profile of feed and how much forage a horse eats, are still areas that require additional education. The reasons behind the uptake or otherwise of available education require further research. Forage producers should be encouraged to regularly provide analyses for their product, so owners have something to help them choose the best complementary feed for their horse. However, as analytical information on forages is highly variable, a consensus of analytical profiles needs to be done and accompanied by education for owners on interpretation of this information. This study has revealed that there are still improvements that could be made in the current feeding practices, and continually monitoring the challenges owners face in making such improvements is necessary if recommendations are to be followed.

## Figures and Tables

**Table 1 animals-13-01280-t001:** Education status of Group 1 (15–40) and Group 2 (41+) horse owners in UK. RC = Riding Club qualifications; BHS = British Horse Society qualifications.

Age	University	Formal (e.g., BHS/RC)	Industry	Totals	Sig
**15–40**	118 ^a^	104 ^c^	72 ^c^	294	
**41+**	28 ^b^	92 ^c^	90 ^c^	210	
**Total**	146	196	162	504	0.003

^abc^ Values in the same column with different superscripts are significantly different at *p* < 0.003.

**Table 2 animals-13-01280-t002:** Stabling and turn out regimens of both Group 1 (15–40 y old) (G1) and Group 2 (41+ y old) (G2) UK horse owners.

	Stabled Part of the Day + Night	Stabled All the Time	Never Stabled	Total	Sig
**G1**	191	33	69	293	
**G2**	137	12	61	210	0.060
	328 ^a^ (65%)	45 ^b^ (9%)	130 ^c^ (26%)	503	0.001

^abc^ Values in the same row not sharing common superscripts are different (*p* < 0.001).

**Table 3 animals-13-01280-t003:** Responses between G1 (15–40 y old) and G2 (40+ y old) to the question ‘Do you stick to a particular brand of fibre-rich feed (i.e., a feed with >18% crude fibre)?

	Yes	No	Totals	Sig
**G1**	205 ^a^	89 ^b^	394	
**G2**	168	42	210	0.010
	373 ^a^	131 ^b^		0.001

^ab^ Values in the same row not sharing letters are sig different (*p* < 0.01).

**Table 4 animals-13-01280-t004:** The number of owners that consistently fed a particular brand of complementary feed/cereal-containing mixed feed.

	Yes	No	Sig
**G1**	166	128	
**G2**	130	80	0.221

**Table 5 animals-13-01280-t005:** Response to the question ‘Do you feel your feeding practices have changed in the last 10 years?

	Last 5 Years	Last 5–10 Years	Last 10+ Years	Not Changed	Sig
**G1 (15–40 y old)**	124 ^a^	31 ^b^	12 ^b^	127 ^a^	0.006
**G2 (41+ y old)**	67 ^a^	38 ^b^	32 ^b^	73 ^a^	

^ab^ Values in the same row not sharing common superscripts differ significantly (*p* < 0.05).

## Data Availability

Data can be obtained from the corresponding author upon request.

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
