# Peer review of "Where Are We Now? Feeds, Feeding Systems and Current Knowledge of UK Horse Owners When Feeding Haylage to Their Horses"

_animals, 2023, doi:10.3390/ani13081280_

Round 1
Reviewer 1 Report
The manuscript outlines the current feeding habits and knowledge of the UK horse-owning public. The results of this manuscript will enable future research and shape how owner education on feeding and nutrition is implemented. However, some areas in this manuscript could be edited to improve clarity.
Simple summary and abstract: the study aim states the aim was to “determine if research findings are 25 implemented in practice; understand owner’s knowledge of haylage, and identify areas that require additional education.” However, there is no reference to whether research findings are taken into account in the survey results. Please rephrase.
Line 29 and throughout- Lowercase p for p-value
Line 19+33- Suggest changing to “Of those who didn’t feed haylage, 66% were not sure how to feed it, 68% worried about aerobic spoilage and 79% stated the bale size was unsuitable.
Line 21+35-education to owners?
Introduction
Line 83- Remove “To be able”
Line 84- fix reference
Line 76-97- This is not fully relevant to the research aims and surprises the reader when it moves into the aims. Consider condensing this section and providing a better link to the aims.
101-103- Start lines with a capital letter
Line 103- Remove “to”
Materials and methods
It may be important here to state that owners could select more than one outcome for some questions as some results in the results add up to over 100% indicating multiple selections.
Possibly a version of the surveys in the supplementary figures could aid in removing some confusion.
Line 108- Define
Line 146- Lowercase p (and throughout the manuscript)
Results-
Table 1- wasn’t the significance level 0.05, not 0.003?
Line 173- Write the abbreviation in full when starting a sentence.
Line 176- Remove “It is interesting”. This is not language that should be used in a results section.
Line 192- Split this into two sentences.
Line 195 (and throughout)- Refrain from starting a sentence with a number.
Line 197-199- Should this be p<0.05 as this is the threshold stated in the methods?
Line 198- 30% of the 5% of owners that did not feed complementary feed? Please make this clearer
Line 200- The type of forage fed was not influenced by age group with 328 owners (67%) feeding hay, 76 (15%) feeding haylage and 96 (19%) feeding hay and another forage.
Line 205-206- This is not a result.
Line 217-220 Is this the percentage of respondents that did not feed haylage?
Line 231- Define WSC- first occurrence.
Line 236- Place 26% in brackets
Line 249- You use respondents and owners interchangeably. Please be consistent.
Line 298- remove “.”
Discussion
The discussion raises some good points for future research and education for horse owners. However, there are some statements that are not backed by results in this manuscript such as the links of feeding regime to obesity.
In addition, suggestions for owners such as the use of weigh scales, testing forages and heart rate monitors are not always plausible. Although these would be great tools, there needs to be an acknowledgement that these are not always viable due to the cost and access to these tools.
Line 330-337- This paragraph could be developed a bit more to discuss reasons for these results rather than just repeat the result. Possibly merge with the following paragraph.
Line 344- This sentence is all results and is also partially stated in the previous paragraph.
Line 349- Without quantification of how much of these feeds are being fed and the management of these horses it is bold to state that this is “concerning”.
Line 371-380- Although the ability for owners to weigh and body condition their horses in order to make management changes accordingly is easy to do, it is also important to acknowledge that weighing horses is not always practical due to cost and logistical constraints. In addition, the use of heart rate monitors comes at a cost to the owner and is not always a practical option, particularly for the leisure riding demographic.
Line 381- Capital letter in the title.
Line 390- Change “1” to “one”.
Line 402-405. This is a result.
Line 420- Remove (NSC).
Lines 442-447- Capital letters and full stops are needed on each line.
Line 452- Comma after choice
Line 454- What was the 86% of?
Other comments
Please remove unnecessary double spaces throughout the manuscript.
It would be helpful to define balancer and complimentary feed as these terms differ between countries. It is stated partway through the discussion but should be earlier in the manuscript.
Reviewer 2 Report
Summary:
1. Line 15: Since the two surveys (S1 & S2) are first mentioned in the passage, it would be better to explain what they are about respectively before interpreting the results separately.
2. Line 16-17: The two percentage rates of those who fed only hay and both hay and haylage seem to be contradictory, same as Line 30-31 in the abstract.
3. Line 18: The “Forage” should be “forage”.
Keywords:
1. Add keywords about horse owner education.
Materials and methods:
1. It would be better to provide the full text of the questionnaires as supplementary materials.
2. Line 139-141: Why is the age of 40 used as the grouping basis? Experience is not a reliable basis.
3. Line 142-143: How to tell if these questions are not affected by age?
Results:
1. Line 149: The two ratios (73% and 26%) should add up to 100%.
2. Line 158-159: It needs to be clarified how you get 75%.
3. Line 166, 279, 282, 299: Please add the percentages in the tables.
4. Line 178: Please add the data in the column “total” in table 2.
5. Line 191: The “SI” should be “S1”.
6. Line 191-192: The three ratios (72%, 19% and 8%) should add up to 100%.
7. Line 200-201: The three ratios (67%, 15% and 19%) should add up to 100%.
8. Line 233-238: The second paragraph in “3.6.” is more about “forage choice” in “3.5.” rather than “owner knowledge” in “3.6.”. It may be more suitable to be put in “3.5.”.
9. Line 285: The “0wners” should be “Owners”.
10. The description is too trivial. It will be more intuitive if the original text of the questionnaire can be attached and the data can be marked on each option of the questionnaire.
Discussion:
1. Line 390-391: Lack of evidence for the reason (to improve the horse-human relationship) for giving 1 bucket feed per day.
2. Line 442-447: Change to a serial number that is not easily confused with the subtitle serial number.
3. Line 469-472: Provide reference about “high level of variability of DM, acidity and nutritional profile of UK haylages…”.
4. Line 356: The format insertion of references in the article is not uniform.
5. Line 475-477: Discussion about the effect of advice-seeking on horses lacks references and looks like speculation.
6. The discussion is not sufficient. Please discuss the difference between the results of this study and previous similar studies, and discuss the changes of feeding in recent years.
7. It is a vital conclusion which is mentioned many times that the horse owners’ perception of haylage is not enough and should be further educated. But the evidence supporting this conclusion, i.e., the real properties of haylage and the owners’ perceptions, is scattered throughout the passage. It would be better to discuss it centrally. E.g., describe haylage’s properties and based on this, which perceptions of the owners are correct, which are imprecise, and which are complete misperceptions.
Conclusion:
1. Line 492-495: The deep-seated reasons for the lack of education, forage analyses, and so on should also be the direction of future research.
Reference:
1. The format of the references needs to be carefully checked and revised per the guidelines.
Reviewer 3 Report
Dear authors, thank you so much for your well-designed and presented work. I really enjoy to read your manuscript and I think that it should be considered for publication.
I just have some comments:
Pay attention to stylistic errors: double space (e.g. line 19, line 33, line 34, line 42); capital letter after “;” (e.g. line 18)… please, make a check throughout the manuscript
Line 15. It is not clear in the simple summary what it is S2
Line 32. …didn’t see the need… to perform forage analyses?
Generally, the introduction is well-written by setting adequate background to introduce the aim of your study. Just some comments:
Line 45. I suggest toad this reference: https://doi.org/10.1016/j.jveb.2021.01.006
Line 50. I apologize but what do you mean by a strong survey by King in 2012? Please define it better or specify
Line 52. It Could be useful to provide deeper discussion and some examples of clinical and behavioural issues (I could strongly suggest Colombino et al. BMC Veterinary Research (2022) 18:338 https://doi.org/10.1186/s12917-022-03433-y ; Raspa et al. BMC Veterinary Research (2022) 18:187 https://doi.org/10.1186/s12917-022-03289-2); Cavallini et al 2022 (10.1016/j.jevs.2022.103940 ); Raspa et al., 2022 Animals (10.3390/ani12141740 ); Rsapa et al 2020 (10.3390/ani10081334))
Line 86. Italics not necessary. Check also other references (line 306, 307 and so on)
Also, Isuggest to add some information regarding the importance of the palatability of the feed, please cite: Vinassa et al 2020 (10.1016/j.applanim.2020.105110)
Line 105. It is not clear if informed consent to the use of data from owners was requested or not...please, add a statement here
Line 108. Could you please describe the acronyms?
Line 109. Surveys were spread by Google form? Or which other platform?
Did you ask about the advices from veterinarians or you assumed that the owners planned the diets for their horses by themselves?
Line 147. Responses or respondents? Please, check 73+26 = 99%
Please include n numbers as well as percentages if possible in a specific Table related to lines 147- 165
Line 155-156. It is not clear the reason why survey 2 did not categorise respondents by age? Consider it acceptable to specify it.
Table 1, 2, 3, 4 and 5. I would suggest to add the n numbers of G1 and G2 respondents, plus add the % (for example: 191, xx%)
Line 191. S1?
Sections 3.5 and 3.6. It could help the readers to organise those results in a Table by describing n numbers and %
Line 285. Please replace the “0” with “O”
Line 338-340. Please reword this sentence, it can be improved.
Line 356. Formatting error in the reference. Please check also in the references list because it is missing
Conclusions. Consider if it could be useful to add a statement regarding the role of veterinarians to help in education.
Reference 5. Please check – I think you should add “conserved”
Round 2
Reviewer 3 Report
Dear authors, after reading the changes to the manuscript I have some concerns regarding some of them not being addressed. Please see my specific comments in the attached file. Best
